# Easy Express Extraction (Triple*E*)—A Universal, Electricity-Free Nucleic Acid Extraction System for the Lab and the Pen

**DOI:** 10.3390/microorganisms10051074

**Published:** 2022-05-23

**Authors:** Christian Korthase, Ahmed Elnagar, Martin Beer, Bernd Hoffmann

**Affiliations:** Institute of Diagnostic Virology, Friedrich-Loeffler-Institut, 17493 Greifswald-Insel Riems, Germany; christian.korthase@fli.de (C.K.); ahmed.elnagar@fli.de (A.E.); martin.beer@fli.de (M.B.)

**Keywords:** nucleic acid extraction, field application, African swine fever virus, lumpy skin disease virus, peste des petits ruminants virus, bluetongue virus

## Abstract

The complexity of the current nucleic acid isolation methods limits their use outside of the modern laboratory environment. Here, we describe a fast and affordable method (easy express extraction, called Triple*E*) as a centrifugation-free and electricity-free nucleic acid isolation method. The procedure is based on the well-established magnetic-bead extraction technology using an in-house self-made magnetic 8-channel and a rod cover. With this extraction system, nucleic acids can be isolated with two simple and universal protocols. One method was designed for the extraction of the nucleic acid in resource-limited “easy labs”, and the other method can be used for RNA/DNA extraction in the field for so-called molecular “pen-side tests”. In both scenarios, users can extract up to 8 samples in 6 to 10 min, without the need for any electricity, centrifuges or robotic systems. In order to evaluate and compare both methods, clinical samples from various viruses (African swine fever virus; lumpy skin disease virus; peste des petits ruminants virus; bluetongue virus), matrices and animals were tested and compared with standard magnetic-bead nucleic acid extraction technology based on the KingFisher platform. Hence, validation data were generated by evaluating two DNA viruses as well as one single-stranded and one double-stranded RNA virus. The results showed that the fast, easy, portable and electricity-free extraction protocols allowed rapid and reliable nucleic acid extraction for a variety of viruses and most likely also for other pathogens, without a substantial loss of sensitivity compared to standard procedures. The speed and simplicity of the methods make them ideally suited for molecular applications, both within and outside the laboratory, including limited-resource settings.

## 1. Introduction

Transboundary animal diseases, such as African swine fever (ASF), bluetongue disease, lumpy skin disease and peste des petits ruminants (PPR), result in serious socio-economic consequences for affected countries [1,2,3,4,5,6,7,8]. Thus, early diagnosis and reaction to disease outbreaks are essential to carry out control activities. Rapid and reliable diagnostic tools are of paramount importance for the confirmation of clinical cases and the early implementation of control measures, which is crucial to prevent further spread of the disease [9].

The diagnosis of the above-mentioned infectious diseases can be performed by direct and/or indirect detection of the infectious agents. The molecular diagnostics by polymerase chain reaction (PCR), isothermal amplification, nucleic acid sequence-based amplification or loop-mediated isothermal amplification are widely used as direct detection methods, as listed in the official World Organisation for Animal Health (OIE) manual of diagnostics [10,11,12]. They are all based on the amplification and detection of viral nucleic acids, so that no pathogens need to be cultivated, and, at the same time, they allow for the relatively rapid confirmation of the disease [13,14,15,16,17,18,19,20,21].

Nonetheless, all of these methods require reliable and safe nucleic acid purification. These can be either manually or automatically performed. The advantages of automated nucleic acid extraction are the streamlined and efficient extraction process, the possibility of high and scalable throughput, the reduced manual handling with a lower risk of contaminations and a less time-consuming extraction work flow. Therefore, although most laboratories rely on manual DNA or RNA extraction methods, automated nucleic acid extraction has become an attractive alternative to labor-intensive manual methods [22,23,24]. For this reason, a growing number of automated extraction platforms are available, and there are multiple reports of their use in assays for pathogen detection [25]. In addition to silica-membrane-based column systems, which realize the flow of the sample to be extracted via centrifugation or the application of vacuum, magnetic-bead-based extraction methods have become more and more widely used. In general, all these systems are designed for the extraction of RNA and DNA, and which system to use in a laboratory depends on many factors (number of samples, sample continuity, matrix, human resources, technical equipment, etc.). A trend towards the use of ready-to-use extraction kits and prefilled systems has been observed in recent years. This further harmonizes and standardizes molecular diagnostics. In addition, such standardized kits will speed up the extraction procedure and will reduce the risk of sample contaminations. The efficiency of these kits has already been showed by Schlottau et al. [26], wherein it was described that, in approximately 20 min, up to 96 samples can be easily extracted without cross contamination between samples [27].

Nevertheless, automated extraction systems are often expensive and not available for all laboratories worldwide. In addition, automatic, as well as manual, nucleic acid extraction needs electricity, at least for repeated centrifugation steps. Thus, the development of an affordable, nonelectric device would cover a large diagnostic market area, since underdeveloped countries and low-budget diagnostic laboratories would profit from low-cost but high-quality extraction methods. Furthermore, electricity-free and rapid nucleic acid extraction is a prerequisite for the realization of so-called “molecular pen-side tests”, which will combine the sensitivity and specificity of molecular diagnostic tests with the simplicity and speed of antigen-based point-of-care test systems.

In our study, we aimed to establish and validate a rapid, reliable, portable and affordable nucleic acid extraction method, called Triple*E*, which does not require extensive technical skills. Therefore, all the listed advantages make this small and portable device a piece of reliable equipment that can be used in the field as a part of a fast-molecular diagnostic tool. Therefore, it can be used both inside and outside the laboratory, as it can be easily adapted to a wide range of downstream molecular assays. It is considered a simple instrument for nucleic acid extraction without the need for centrifugation steps, which is a significant advantage of this technique over other commercial extraction systems. [28]. For an initial validation, samples of different matrices infected with four viruses of emerging diseases were used. Nucleic acid of two DNA viruses (African swine fever virus (ASFV), lumpy skin disease virus (LSDV)) and two RNA viruses (peste des petits ruminants virus (PPRV) and bluetongue virus (BTV)) were extracted with the novel Triple*E* procedure in comparison to a standard magnetic bead extraction method on an semi-automated KingFisher platform and analyzed in well-established follow-up real-time PCR systems.

## 2. Materials and Methods

### 2.1. Sample Collection and Viruses

A panel composed of 64 samples was used. (i) ASFV-positive samples (*n* = 16) consisted of field-collected specimens (*n* = 12; EDTA wild boar blood samples) that had been submitted to the Friedrich-Loeffler-Institut for disease confirmation and samples (*n* = 4; EDTA blood samples) obtained from experimentally infected domestic pigs. (ii) LSDV-positive samples (*n* = 16) were composed of EDTA blood, serum, oral/nasal swabs and crusted skin specimens collected from experimentally infected ruminants. (iii) PPRV-positive samples (*n* = 16) were composed of oronasal and conjunctival swabs and spleens from experimentally infected goats. (iv) BTV-positive samples (*n* = 16) were composed of EDTA blood samples from experimentally infected ruminants. The sample composition is listed in detail in Appendix A.

All experimental protocols were reviewed by a state ethics commission and approved by the competent authority (Landesamt für Landwirtschaft, Lebensmittelsicherheit und Fischerei (LALLF) Mecklenburg-Vorpommern, Rostock, Germany). Six different animal trials were conducted at the FLI with the following reference numbers: ASFV Estonia 2014 (M-V/TSD/7221.3-2.011/19), LSDV Macedonia 2016 (M-V/TSD/7221.3-2.1-022/10), LSDV-Nigeria-V281 (M-V/TSD/7221.3-2-004/18), PPRV- Kurdistan 2011 (M-V/TSD/7221.3-1-018/14), BTV-27 and BTV-4 (M-V/TSD/7221.3-1.1-058/10), as well as BTV-33 and BTV-8 (M-V/TSD/7221.3-1-048/19).

### 2.2. Nucleic Acid Extraction

All samples were extracted twice by each of the magnetic bead-based extraction system. A preliminary step was carried out for all swab-collected specimens. In detail, swabs were collected using FLOQSwabs (Copan, Brescia, Italy), submerged into 2 mL of non-supplemented MEM medium, shaken for 30 min at RT and transferred to 2 mL tubes (Eppendorf, Hamburg, Germany). Samples were stored at 4 °C until extraction. A summary of all four validated extraction methods can be found in the Appendix A.

#### 2.2.1. KingFisher Flex Extraction System

As standard reference method, the NucleoMagVet kit (Macherey-Nagel, Düren, Germany) on the semi-automated KingFisher Flex platform (Thermo-Fisher-Scientific, Waltham, MA, USA) was applied. Extraction was performed following the manufacturer’s instructions. Briefly, 100 µL sample volume was added to the KingFisher 96 deep-well plate, followed by the addition of 20 µL Proteinase K and 100 µL lysis buffer VL1. Subsequently, 350 µL binding buffer VEB and 20 µL NucleoMag B-Beads were added to the sample-lysis buffer mix. After three washing steps, the extracted nucleic acids were eluted in 100 µL elution buffer VEL.

#### 2.2.2. IndiMag 48 Extraction System

A second method was used for comparative purposes, the IndiMag Pathogen Kit on the IndiMag 48 platform was used (Indical Bioscience, Leipzig, Germany), following the instructions described by Elnagar et al. [29].

#### 2.2.3. Easy Express Extraction (Triple*E*) System

Finally, the Triple*E* method was established and validated. This extraction system was composed by a nonelectric extraction procedure using the IndiMag Pathogen Kit (Indical Bioscience).

##### Extraction Instrument

As shown in Figure 1, the Triple*E* procedure was performed by using three hardware components. The first one was the in-house self-made magnetic 8-channel (Figure 1A) which originated from an IndiMag 48 extraction machine and was subsequently modified for manual handling. The second was the IndiMag 48 PW Rod cover as shown in Figure 1B, which was used in combination with the magnetic 8-channel (Figure 1C); finally, the third was an IndiMag 48 PW 24-Sample Block (Figure 1D), wherein the buffers were placed. All three listed components were obtained from Indical Biosciences (Leipzig, Germany).

##### Extraction Plate and Buffers

Before the extraction could be started, plates were prepared as follows. IndiMag Pathogen Kit buffers were placed in the IndiMag 48 PW 24-Sample Block (96-deep well plate) as described in Figure 2. Briefly, in column 1, the proteinase K (20 µL/well) was placed. In column 2, the first washing buffer AW1 (500 µL/well) mixed with the magnetic beads (25 µL MagAttract suspension G/well) were located. In columns 3, 4 and 5, three further washing buffers (AW1, AW2 and ethanol (80%)) were placed (500 µL/well). In column 7, the AVE elution buffer (100 µL/well) was filled. The separate location of the elution buffer should minimize the risk of contamination during the extraction procedure. Subsequently, the prefilled 96-deep well plate was covered with a Thermo-Bond Heat Seal foil (Biozym Scientific, Oldendorf, Germany) and heat-sealed for 4 s at 175 °C using the Sally Heat Sealer (Biozym Scientific). Prefilled plates were stored at room temperature (RT) until further use. 

##### Extraction Workflow

Next, the Triple*E* easy-lab workflow was carried out. To this end, the in-house self-made magnetic 8-channel and the rod cover were used either separated or combined during the extraction, depending on the requirements of each step, as illustrated in Figure 3. The following extraction protocol was carried out:
*Lysis-binding steps*: A 100 µL sample was placed on 1.5 mL tubes (Eppendorf) prefilled with 100 µL VXL lysis buffer and 400 µL ACB binding buffer (IndiMag Pathogen Kit, Indical Bioscience). Then the 600 µL sample-lysis-binding mix was thoroughly mixed by repeated pipetting and added to the first column (including the Proteinase K) of the prefilled 96-deep well plate.Then, magnetic beads were collected from column 2 with the magnetic channel inserted into the rod cover. This was carried out by dipping up and down the magnetic channel-rod cover up to 10 times (Figure 3—1). Subsequently, the magnetic channel-rod cover with the attached magnetic beads was transferred into column 1 (Figure 3—2), then the magnetic channel was removed and placed the parking position in column 12. Now, the separate rod cover in column 1 was dipped up and down 30 times and was incubated for 3 min at RT (Figure 3—3). Next, the magnetic channel, picked up from the park position, was inserted into the rod cover (Figure 3—4), and the combo was dipped slowly 10 times up and down to collect the magnetic beads again.*Washing steps*: The magnetic channel-rod cover with the attached magnetic beads was inserted into column 2 (Figure 3—5). The washing step was performed by dipping up and down 30 times with the combined magnetic channel-rod cover without the complete releasing of the magnetic beads. Detached beads were recollected by dipping with slower movements 10 times. This latter step was used to catch the maximum number of magnetic beads free in solution. Subsequently, the described washing procedure was applied to the next three washing steps using columns 3, 4 and 5 (Figure 3—6–8).*Elution step*: Finally, the magnetic channel-rod cover with the attached magnetic beads was inserted into column 7 (Figure 3—9) and was dipped up and down 30 times, again followed by a dipping step consisting of 10 slower movements for catching the maximum number of magnetic beads. Thereafter, the rod cover and the attached magnetic beads were discarded. The ready-to-use nucleic acids remained in column 7 for subsequent real-time PCR amplification or other molecular analyses.

The Triple*E* point-of-care (POC) protocol was performed with modified incubation and dipping steps as follows: (i) the incubation time for the binding step was reduced from 3 min to 1 min, (ii) the fast up and down dippings during the washing steps were reduced from 30 to 10 and (iii) the up and down dippings for magnetic bead collection with slower movements were also reduced from 10 to 5 times. All reagents and volumes were applied as described in the Triple*E* easy-lab protocol.

### 2.3. Real-Time PCR

All primers and probes used in this study are listed in Table 1. The FAM mix consisted of: 10 µL of each primer (100 pmol/µL), 2.5 µL probe (100 pmol/µL) and 77.5 µL 0.1 × TE buffer (pH 8.0). The HEX ß-Actin mix was composed of 2.5 µL of each primer (100 pmol/µL), 2.5 µL probe (100 pmol/µL) and 77.5 µL 0.1× TE buffer (pH 8.0).

PerfeCTa qPCR ToughMix (Quanta BioSciences, Gaithersburg, MD, USA) was used for ASFV and LSDV nucleic acid amplification. The reaction mix consisted 1.75 µL nuclease-free water, 6.25 µL PerfeCTa qPCR ToughMix, 1 µL FAM mix, 1 µL HEX ß Actin mix and 2.5 µL DNA template. The temperature profile was 3 min activation of Taq polymerase at 95 °C, followed by 45 cycles of 15 s at 95 °C denaturation, 15 s at 60 °C annealing and 15 s at 72 °C elongation. [30,31,32]. Additionally, RT-qPCR for BTV and PPRV was run with qScript XLT One-Step RT-qPCR ToughMix (Quanta BioSciences, Gaithersburg, MD, USA). Before each RT-qPCR was performed, denaturation of the double-stranded extracted RNA was carried out as previously described [33]. The reaction mix was composed of 1.75 μL nuclease-free water, 6.25 μL qScript XLT One-Step RT-qPCR ToughMix, 1 µL FAM mix, 1 µL HEX ß Actin mix and 2.5 µL RNA template. The temperature profile was 10 min reverse transcription at 50 °C, 1 min activation of Taq polymerase at 95 °C and 45 cycles 15 s at 95 °C denaturation, 20 s at 57 °C annealing and 30 s at 72 °C elongation. All analyses were measured with the CFX96 Real-Time System (BIO-RAD, Hercules, USA). Fluorescence data were collected during the annealing phase, and results were considered positive when Ct values were <45.
microorganisms-10-01074-t001_Table 1Table 1Sequences of primers and probes.PCR AssayGenomeDetection ofPrimer/ProbeSequence 5′-3′Amplicon(Base Pair)ReferenceASFV-P72-IVI-mixASFVASFV-p72IVI-FGAT GAT GAT TAC CTT YGC TTT GAA78Haines et al., 2013 [30]ASFV-p72IVI-RTCT CTT GCT CTR GAT ACR TTA ATA TGAASFV-p72IVI-FAMFAM-CCA CGG GAG GAA TAC CAA CCC AGT G-BHQ1Capri-p32-mixCapripoxvirusCapri-p32forAAA ACG GTA TAT GGA ATA GAG TTG GAA89Bowden et al., 2008 [31]modified;Dietze et al., 2018 [32]Capri-p32revAAA TGA AAC CAA TGG ATG GGA TACapri-p32-FAMFAM-ATG GAT GGC TCA TAG ATT TCC TGA T-BHQ1Pan BTV-IVI-mixBTVOrru_BTV_IVI_F2TGG AYA AAG CRA TGT CAA A97OIE terrestrial manual (version May 2021)Orru_BTV_IVI_R2ACR TCA TCA CGA AAC GCT TCOrru_BTV_IVI_FAMFAM-ARG CTG CAT TCG CAT CGT ACG C-BHQ1PPRV-Batten-mixPPRVPPRV-N-483FAGA GTT CAA TAT GTT RTT AGC CTC CAT142Batten et al., 2011 [34]PPRV-N-624RTTC CCC ART CAC TCT YCT TTG TPPRV-N-551FAMFAM-CAC CGG AYA CKG CAG CTG ACT CAG AA-BHQ1ß-Actin-DNA-mix 2beta-actin mRNAACT-1030-FAGC GCA AGT ACT CCG TGT G106Toussaint et al., 2007 [35] modified; Wernike et al., 2011 [36]ACT-1135-RCGG ACT CAT CGT ACT CCT GCT TACT-1081-HEXHEX-TCG CTG TCC ACC TTC CAG CAG ATG T-BHQ1

### 2.4. Data Analyses and Statistics

Data were recorded and evaluated using Microsoft Excel 2019 (Microsoft Deutschland GmbH, Munich, Germany). The analytical performance of each extraction method was carried out by comparing the average differences of Ct values using the Bland–Altman test [37]. To this end, this test considers the two extraction systems to be in agreement, if their results fall within the so-called limit of agreement (LoA) interval. This interval was calculated using the mean difference ± 1.96 standard deviation (SD) of the Ct values obtained using both extraction systems.

To test the analytical sensitivity of each method, ten-fold dilution series of virus-positive samples were prepared. For each extraction method, the samples were tested in duplicates and analyzed with the standard (RT)-qPCR assay as described. PCR efficiencies were calculated based on the resulting standard curves. Next, regression analysis and the Pearson’s correlation coefficient were calculated to compare the extraction.

GraphPad Prism 9 (Graphpad Software Inc., San Diego, CA, USA) was used for statistical analyses and graph creation.

## 3. Results

### 3.1. Reproducibility of the Extraction Methods

A panel of field- and laboratory-collected samples was used to assess the diagnostic sensitivity of the rapid extraction protocols for ASFV, PPRV, BTV and LSDV. As presented in Table 2, mean Ct values, as well as the intra-run variability of the different extraction systems, showed that all studied methods maintained high reproducibility. The automated Triple*E* extraction instrument delivered comparable results as the commercially available alternatives for the DNA viruses. However, after the extraction and amplification of BTV and PPRV, a difference of approximately 2 Ct values was observed between the automated and manual Triple*E* systems.

### 3.2. Analytical Performance of the Extraction Methods

The agreement between the KingFisher Flex automated extraction and each of the other used methods was evaluated using Bland–Altman plots (see Figure 4). All recorded row data are summarized in Appendix A.

When comparing the ASFV results, a point-by-point comparison showed a low degree of variability with a trend for strong detection in all samples for all devices. Nonetheless, for β-actin detection the bias, i.e., the average discrepancy that could indicate a systematic difference, was highest for the IndiMag 48 extraction method. This observation coincided with wide limits of agreement and several samples outside the limits (see Figure 4).

The LSDV detection was in general successful for all samples using all four methods. Only one sample was borderline detected using the Triple*E* methods, because only three of four results were positive in the qPCR. Furthermore, a wider limit of agreement and some samples outside the limits for Triple*E* systems could be ascertained (see Figure 4). A manual in-detail comparison revealed that samples obtained from swabs were detected with a distinct shift in Ct values for the extraction of the virus. Similarly, the detection of β-actin in these samples showed a wider LoA for Triple*E* methods when compared with IndiMag 48.

Subsequently, BTV-positive blood displayed a very similar outcome when comparing all extraction systems. Samples above a Ct value of 30 exceed the LoA, suggesting that samples up to these values are in agreement and that the results obtained by any of these extraction methods are comparable. In addition, the β-actin results always lay between the LoA in all systems with almost no discrepancies.

Finally, for PPRV, the Bland–Altman test showed a perfect match, with no LoA exceeded for either the target or the endogenous internal control. Overall, all systems and viruses revealed that, for all sample matrices, with the exception of LSDV-infected swabs, all methods are in agreement with the KingFisher Flex automated method. Hence, no under- or over-estimation of Ct values was detected in this study for any of the hereby tested methods.

### 3.3. Linearity and Analytical Sensitivity of Extraction Methods

Finally, the analytical sensitivity of the four extraction methods was tested for each virus by performing a ten-fold dilution series of virus-positive samples. Overall, all extraction systems allowed viral detection up to a 10^−4^ dilution. Linearity for ASFV and PPRV indicated a 0.99 correlation coefficient, while LSDV and BTV ranged from 0.83 to 0.97 for the Triple*E* system (see Figure 5).

## 4. Discussion

Animal disease outbreaks of ASF, LSD, bluetongue and PPR have caused suffering, death and economic losses worldwide [1,3,5,6,7,38,39]. These viral pathogens can threaten global health and food security. Nucleic-acid-based diagnosis has allowed authorities to rapidly react and control the outbreaks; thus, (RT)-qPCR has become the standard molecular-diagnostic tool in many countries [40]. However, nucleic acid extraction remains one of the most important steps leading to a successful diagnosis. Over the past few years, significant progress has been made to simplify and speed up the viral nucleic acid isolation process of different sample matrices [9,26,29,41] Therefore, in this study, three extraction systems were compared with our well-established reference method, the NucleoMagVet kit (Macherey-Nagel) on the KingFisher Flex platform. Beside an alternative automated extraction method, the commercially available IndiMag Pathogen Kit on the IndiMag 48 platform (both from Indical Bioscience, Leipzig, Germany), and two manual extraction systems based on magnetic bead technology, were evaluated for the simple, universal and electricity-free extraction of nucleic acid (called “Triple*E*” (easy express extraction).

Both manual extraction methods use a hand-made, magnetic channel for processing eight samples in parallel. This 8-well magnetic channel, as well as the rod cover and the deep well plate, originate from the IndiMag 48 system. In addition, the reagents of the IndiMag Pathogen Kit were implemented in the manual systems. Differences exist regarding the manual processing of the magnetic beads and two additional washing steps added in the universal, electricity-free extraction procedure. Both of these additional washing steps are not possible for the automated IndiMag 48 systems based on technical limitations. The processing of the magnetic beads by hand overcome this technical limitation and allow the integration of several additional washing steps in the extraction process. Here, we included an additional washing step using the AW1 buffer and a second additional washing step using 80% ethanol for the further reduction of inhibitory factors in the eluate. For the simplification of sample processing, the magnetic channel was not removed from the rod cover during the washing steps. The limited washing effect by the non-released magnetic beads was widely compensated for by the two additional washing steps.

The described both Triple*E* systems (easy-lab and POC) can be also defined as two variants of one extraction method. For easier understanding, both variants/methods were described and analyzed separately in the presented study. The incubation time for lysis and binding, as well as the number of movements for bead washing and bead collection, are different between both methods/variants. The “easy lab” system is recommended for resource-limited labs without the possibility of using a robotic system. The “pen-side” system is further speed-optimized, and the proposed application is molecular testing in the field (pen-side, point-of-care). The 96-well plate can be prepared and stored for a longer time at room temperature. Accordingly, prefilled extraction plates for IndiMag 48 or the KingFisher platform based on the IndiMag Pathogen kit will be offered by Indical Biosciences. Thus, it can be concluded that prefilled Triple*E* plates will be functional for long-term storage and applications in the field.

Samples from a wide range of matrices, hosts and viruses were used for the comparative validation approach, and the NucleoMagVet kit on the KingFisher platform acted as reference extraction system. The functionality and power of the alternative automated system (IndiMag Pathogen kit on the IndiMag 48 robotic platform) for ASFV could be shown previously [29]. Here, we present similar validation data for additional viral pathogens, namely LSDV, BTV and PPRV. The chosen viruses reflect a broad range of types of viral nucleic acid (dsDNA, dsRNA and ssRNA), and, based on the analyzed data, the suitability of the IndiMag system for the genome extraction of different viral genomes could be confirmed. Thus, the IndiMag system represents a useful and practicable alternative automated extraction system, especially if 48 or fewer samples are processed in parallel.

Nevertheless, both automated systems need the robotic platforms and electricity for extraction. For some labs with limited resources, unstable supply of electricity or for molecular analyses in the field, the application of robotic systems can be problematic or impossible. In particular, the further development of molecular pen-side tests requires techniques that are mobile and independent from public electricity. For PCR and isothermal amplification, devices with an integrated battery are still available. An example for such techniques are the Franklin cycler from Biomeme, the Liberty16 from Ubiquitome or the Genie III from Optigene. The simple and mobile, electricity-free or battery-based extraction of nucleic acid is still a bottleneck for the further improvement and acceptance of a molecular pen-side test. Here, the M1 sample preparation cartridge from Biomeme represents a commercially available electricity-free nucleic acid extraction systems, which has been successful applied for molecular pen-side tests of SARS-CoV-2 and other pathogens [42,43]. Nevertheless, the M1 sample preparation cartridge needs 5–10 min for one sample and showed reduced sensitivity for protein-rich samples like serum or blood [43]. Nucleic acid extraction with alternative, but electricity-based, field methods are also described [44,45]. Nevertheless, our Triple*E* systems allow the extraction of up to 8 liquid samples in 5–10 min, independent from the sample matrix, host and most likely the pathogen. The chemical basis is the well-established and FDA-approved IndiMag Pathogen kit, which was developed for the viral RNA/DNA and bacterial DNA extraction from a broad range of veterinary sample matrices, e.g., whole blood, serum, swabs and other body fluids. On the other side, the flexible Triple*E* hardware is universal for magnetic bead processing. Thus, the use of alternative extraction kits (e.g., optimized for food or environmental samples) is in general possible, but the incubation times and magnetic bead dipping movements should be validated accordingly for maximum sensitivity. Ready-to-use reagents and buffers minimize human error and lead to further harmonized and standardized nucleic acid extraction. The kits used in this study ensured reliable and robust DNA and RNA extraction, which is advantageous for laboratories with an adequate budget. Nevertheless, the extraction procedures presented here are open and flexible, and thus the use of inexpensive, homemade nucleic acid extraction buffers is also possible [46].

The presented data showed that the Triple*E* systems, easy-lab and POC could successfully isolate viral DNA and RNA from the four targeted viruses. Qualitatively, all tested samples were successfully detected by PCR, and a good agreement of Ct values between the automated and the manual systems was observed. 

NucleoMagVet kit as highlighted by the Bland–Altmann test [37]. However, according to the data presented in this study, one matrix seemed to influence the LSDV positivity of the qPCR results. Swabs obtained from experimentally infected ruminants showed a shift in Ct values for sample detection that exceeded the LoA. The reason for the shift of approximately 2 Ct values for the extraction with the Triple*E* system of LSDV-positive swab samples remains unclear. Nevertheless, for the molecular testing of clinically affected cattle/animals, the reduced sensitivity of 2 Ct values seems acceptable based on the expected high viral genome load in oral and nasal swabs [47].

In terms of reproducibility, the Triple*E* systems showed low intra-run variations when compared with the two automated extraction methods. This fact will be confirmed by a comparison of the data from the internal control assay. Here, the housekeeping gene β-actin was used to ensure the quality of the host genetic material [36]. All automated extraction methods studied here allowed the successful detection of the endogenous internal control in all samples examined. Furthermore, the data of the internal β-actin control of the Triple*E* system show only very slight deviations from the results of the automated extractions. Thus, the functionality of the internal process control is successfully confirmed in the manual procedures, which is of particular importance for virus-negative samples [40,48].

## 5. Conclusions

The automated IndiMag system and the manual Triple*E* provide very comparable results to the NucleoMag/KingFisher system for extraction of viral nucleic acid. The Triple*E* system represents an easy-to-handle manual method for the electricity-free extraction of DNA and RNA in diagnostic laboratories with limited resources. Furthermore, it can easily be implemented in the field for the extraction of nucleic acid using molecular pen-side tests. Our study offers solid data supporting this nonelectric, sensitive and robust extraction method for up to 8 samples in less than 10 min. Direct detection and characterization of pathogens by nucleic-acid-based detection and sequencing techniques will continue to gain importance. The rapid and cost-effective identification of pathogens in endemic, often low-resource countries or directly in the field will play a crucial role for the surveillance and control of animal health worldwide. The Triple*E* system we have developed and evaluated could make a valuable contribution to this.

## Figures and Tables

**Figure 1 microorganisms-10-01074-f001:**
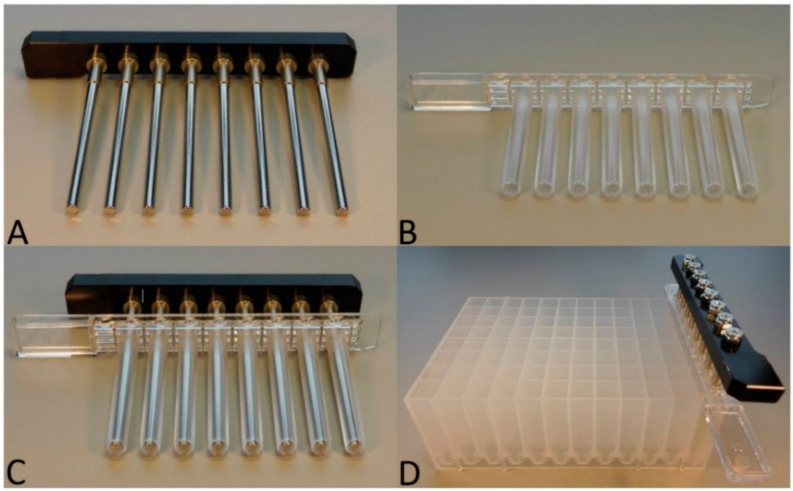
Materials used for the Triple*E* system. (**A**) In-house-self-made magnetic 8-channel. (**B**) IndiMag 48 PW Rod cover. (**C**) Magnetic channel combined with rod cover. (**D**) Magnetic-tip comp inserted into the park position in column 12 of the IndiMag 48 PW 24- Sample Block.

**Figure 2 microorganisms-10-01074-f002:**
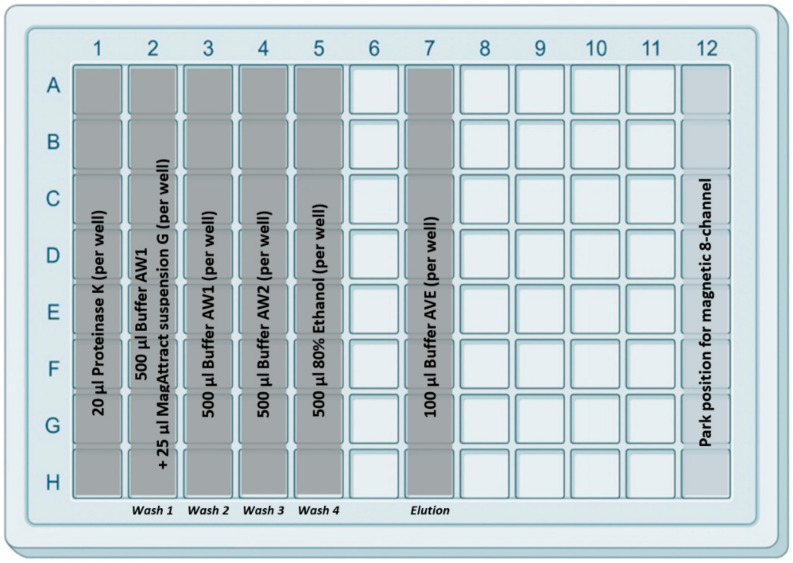
Design of the 96-deep well plate with the prefilled reagents. In the first column, 20 µL proteinase K solution was added, and the second column was filled with 25 µL of magnetic beads (Mag-Attract Suspension G) per well. Washing buffers AW1, AW2 and 80% Ethanol were filled in columns 2 to 5 (500 µL/well) as illustrated. Finally, in column 7, 100 µL/well of elution buffer AVE was added. Columns 6 and 8–12 remained empty. (The figure was created with a 96-well square well plate template from BioRender.com with modifications).

**Figure 3 microorganisms-10-01074-f003:**
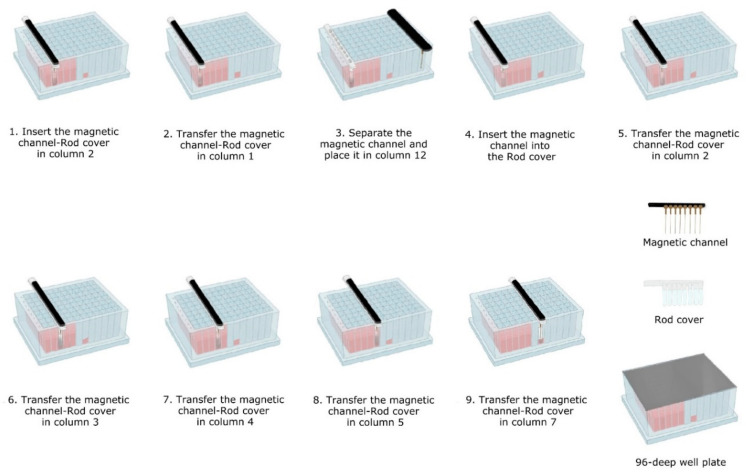
Workflow performed for the Triple*E* easy-lab and point-of-care (POC).

**Figure 4 microorganisms-10-01074-f004:**
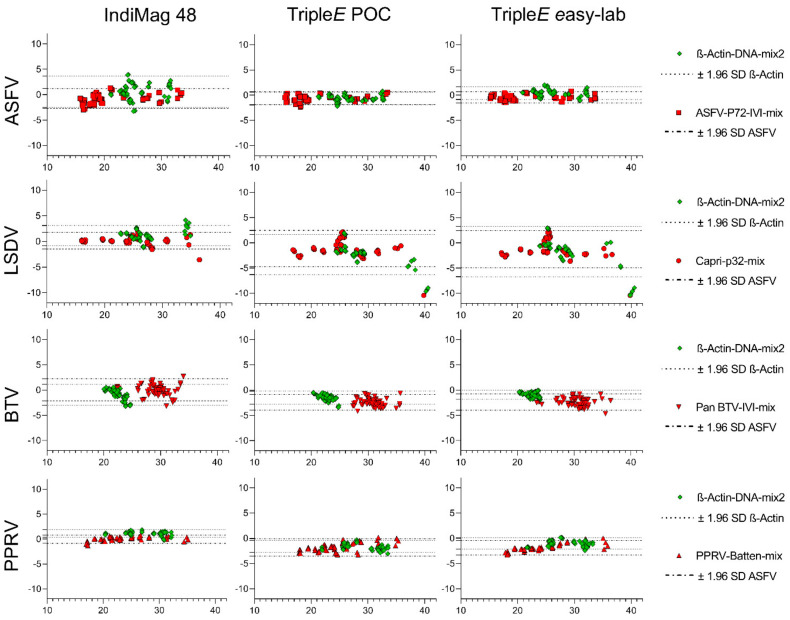
**Evaluation of the analytical performance of the extraction methods.** Bland-Altman plots comparing the KingFisher Flex automated method to the commercially available IndiMag 48 and the hand-made Triple*E* method. The dotted lines represent the limits (upper and lower) of agreement, for both the virus target (red) and the ß-actin (green). The plots show at the Y axis the differences between the Ct values obtained after real-time amplification for each of the evaluated viruses after KingFisher Flex extraction and at the X axis the tested systems against the average of the Ct values detected.

**Figure 5 microorganisms-10-01074-f005:**
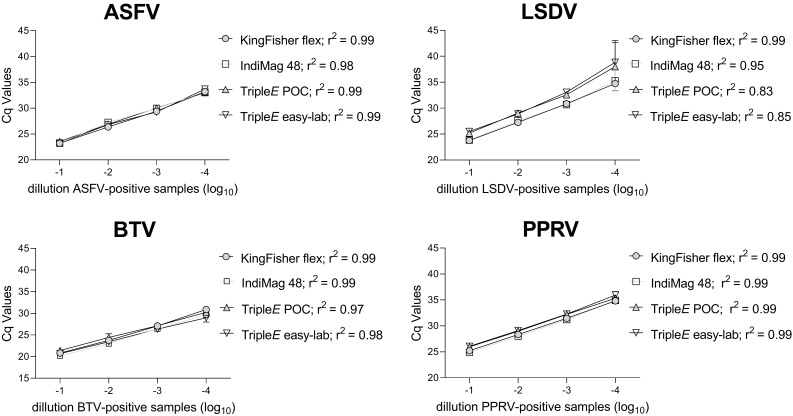
**Linearity and analytical sensitivity of the four different extraction methods.** The ASFV-, LSDV-, PPRV- and BTV-positive samples were diluted ten-fold. Extracted RNA or DNA was quantified as previously described. Regression lines are illustrated and the correlation coefficient given in the legend for each method. POC: Point-of-care.

**Table 2 microorganisms-10-01074-t002:** Reproducibility of the standard protocol compared with four extraction protocols. Intra-run variability.

Virus	Extraction	Mean Ct	SD	CV (%)
ASFV	KingFisher Flex	21.03	0.20	0.92
	IndiMag 48	21.77	0.26	1.18
	Triple*E* POC	21.63	0.18	0.93
	Triple*E* easy-lab	21.47	0.17	0.81
LSDV	KingFisher Flex	24.87	0.16	0.63
	IndiMag 48	24.72	0.27	0.98
	Triple*E* POC	25.82	0.18	0.01
	Triple*E* easy-lab	25.99	0.20	0.72
PPRV	KingFisher Flex	23.87	0.14	0.59
	IndiMag 48	23.44	0.18	0.77
	Triple*E* POC	25.23	0.30	1.13
	Triple*E* easy-lab	25.28	0.15	0.56
BTV	KingFisher Flex	29.14	0.33	1.12
	IndiMag 48	29.07	0.50	1.72
	Triple*E* POC	31.55	0.41	1.29
	Triple*E* easy-lab	31.51	0.51	1.59

SD = standard deviation; CV% = coefficient of variation.

## Data Availability

The data set used and/or analyzed during the current study are available from the corresponding author on reasonable request.

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
