# Peer review of "Easy Express Extraction (TripleE)—A Universal, Electricity-Free Nucleic Acid Extraction System for the Lab and the Pen"

_microorganisms, 2022, doi:10.3390/microorganisms10051074_

Round 1
Reviewer 1 Report
Well written and clearly presented.
I found one type on line 136 "can be started"
Reviewer 2 Report
The paper elucidates a method for the purification of nucleic acids without the need for highly skilled personnel and a well-equipped laboratory.
The manuscript is well written and needs the minor revision herein listed
1-Add ref. for lines 40-43 e.g. the handbooks for molecular biology, protocols in mol. biology
2-lines 48-53. Better elucidate the advantages of automated extraction.
3- lines 53-54. The authors state "Such standardized kits will speed up the extraction procedure 53 and will reduce the risk of sample contaminations"
It would be helpful to the reader, at this point of the text, to have a clear comparison between the time needed for the procedures mentioned as well as the contamination percentage to better clarify the advantages/disadvantages.
4- In the abstract and introduction I would suggest to focus on the portability of the device as main advantage rather then electriciy to maximize the impact
5- Authors don't mention magnetic beads or vacuum methods as centrifugation-free DNA purification methods in the introduction. Relate them to the limits and advantages in comparison with the other techniques
6- Revise the caption of fig.3
7- Increase the readability of fig 4
8- Increase the readability of fig 5
Reviewer 3 Report
In this manuscript, the authors evaluated three extraction systems ((i) IndiMag 48 extraction system, Easy Express Extraction system; (ii) “easy lab” and (iii) “pen-side” protocols), which were compared with well-established reference method, the NucleoMagVet Kit on the KingFisher Flex platform. Overall, the manuscript was well written, and the extraction systems were completely validated with 64 clinical samples infected with either single-stranded or double-stranded RNA viruses. Therefore, I suggest the acceptance of this manuscript after the minor revision.
- It would be better to use different terminologies for “easy lab” and “pen-side”.
- To facilitate the understanding of differences in the experimental procedures, it would be better to prepare the table that summarizes the three extraction systems and reference method.
- The following, up-to-date, related references need to be included to furnish the background information of the manuscript
- doi.org/10.1016/j.snb.2022.131871 & doi.org/10.1016/j.bios.2022.114221 & doi.org/10.1016/j.aca.2022.339781 & doi.org/10.3390/chemosensors9070167
